# Effective Purification of Eutrophic Wastewater from the Beverage Industry by Microbubbles

**Kimio Fukami** [1,*,†] **, Tatsuro Oogi** [1] **, Kohtaro Motomura** [1] **, Tomoka Morita** [2] **, Masaoki Sakamoto** [2] **and Takashi Hata** [3]

[1] Faculty of Agriculture and Marine Science, Kochi University, Nankoku 783-8502, Kochi, Japan; tatsuro0098@gmail.com (T.O.); motocaster07@gmail.com (K.M.)

[2] Sakamoto-Giken Inc., Nankoku 783-0007, Kochi, Japan; s1518@gm.kochi-ct.jp (T.M.); masaoki@sakamotogiken.com (M.S.)

[3] Department of Social Design Engineering, National Institute of Technology (KOSEN), Kochi College, Nankoku 783-8508, Kochi, Japan; thata@ms.kochi-ct.ac.jp

[*] Correspondence: fukami@kochi-u.ac.jp; Tel.: +81-88-843-4864

[†] Present address: Kochi Study Center, The Open University of Japan, Akebono-cho 780-8072, Kochi, Japan.

**Abstract:** Beverage industries often discharge large amounts of organic matter with their wastewater. Purification of the effluent is their obligation, but it is nontrivial. Among wastewater components, removal of dissolved organic matter often requires much effort. Therefore, a special effective technique must be considered. Microbubbles (1–100 μm) have several special properties of relevance to wastewater treatment. In this study, the effectiveness of microbubbles for treating and purifying beverage wastewater was evaluated. Orange juice, lactic acid drink, and milk were used as model substrates of dissolved organic matter, and degradation experiments were carried out. Rates of air supply by microbubbles were 0.05% (air/wastewater) min$^{-1}$. Results indicated that the total organic carbon (TOC) in an experimental vessel containing milk (high nitrogen content) decreased by 93.1% from 11.0 to 0.76 g during a 10-day incubation. The TOC of lactic acid drink (least nitrogen content) decreased by 66.3%, from 15.6 to 5.26 g, and the TOC of orange juice (medium nitrogen content) decreased by 82.7%, from 14.8 to 2.55 g. Large amounts of particulate organic matter floated on the water surface in the milk with microbubbles and were removed easily, while almost no floating materials were observed in the orange juice and lactic acid drink. In contrast, in the macrobubble treatment (diameter 0.1 to 2 mm), only 37.0% of TOC in the milk was removed. Whereas the macrobubble treatments were anaerobic throughout the incubations, the microbubble treatments returned to aerobic conditions quickly, and brought 10 times greater bacterial abundances (>10$^8$ cells mL$^{-1}$). These results suggest that microbubbles are much superior to macrobubbles in supplying oxygen and accelerating the growth of aerobic bacteria, and that wastewater containing more nitrogenous compounds was purified more effectively than that with less nitrogen by microbial degradation and floating separation.

**Keywords:** microbubbles; beverage wastewater; purification; floatation separation; oxygen supply; degradation of organic matter

## 1. Introduction

"Fine bubbles" are defined by the International Organization for Standardization (ISO) as bubbles having diameters of less than 100 μm. Among fine bubbles, those with diameters of 1–100 μm are referred to as microbubbles (MiBs) [1–3]. MiBs have several special properties that include the following: (1) much slower rising velocity compared with macrobubbles (MaBs) (with diameters between 100 μm and 2 mm) [2,4]; (2) large ratio of contact surface area to bubble volume [5]; (3) high internal gas pressure [6–8]; (4) electrically charged bubble surface [7,9]; (5) tendency to shrink and eventually dissolve in water [4]. These characteristics cause MiBs to be stable in the water column longer than





MaBs, and for much of the bubble gas to dissolve into the water. In addition, charged dissolved organic and fine particulate materials tend to be adsorbed, concentrated, and aggregated on microbubble surfaces [8,10,11].

The special properties of MiBs have been applied in many fields [12]. One effective application is to supply dissolved oxygen (DO) to aquaculture fields [13–16]. Srithongouthai et al. [13] reported that use of a system that generates microscopic bubbles (5–40 μm in diameter) allows DO levels in the net pens of fish farms to recover during the night from ~5 mg $L^{-1}$ to saturation levels (>6 mg $L^{-1}$). Joni et al. [17] also reported that the application of MiBs in an aquaculture system increased the DO from 4.5 mg $L^{-1}$ to 7.9 mg $L^{-1}$. In addition to their use in aquaculture facilities, MiBs have been used to improve the water quality of wastewater contaminated with various hazardous materials, such as azo dye [18], methylene blue [19], phenol [6,20], palm oil [8], diesel oil [21], trichloroethylene [22], surfactants [3], pesticides [23], and pathogenic bacteria [24]. MiBs have also been used to remove microalgal cells [25] and planktonic crustaceans [26].

Even if wastewater is not contaminated with hazardous materials, the discharge of wastewater containing an excessive amount of organic matter to the environment can cause serious eutrophication problems and pollute the aquatic environment. The result can be outbreaks of harmful algal blooms [27], oxygen depletion [28], and the mass mortality of aerobic benthic animals [29]. Reduction of the concentration of organic matter in wastewater and the associated organic loading of the aquatic environment is an important and urgent issue [30]. The food and beverage industries, in particular, often discharge relatively large amounts of organic matter with their wastewater [31,32]. Facilities associated with such industries have a responsibility to treat and purify their effluent before releasing it [33]. It is a nontrivial obligation for them to build and operate wastewater treatment facilities [34].

Among the various components in wastewater, particulate matter is relatively easy to remove from the effluent because it can be separated by filtration [35–38], floatation [39], and/or sedimentation [33]. Therefore, removal of sludge and suspended solids from wastewaters is not technically difficult. In contrast, removal of dissolved organic matter (DOM) often requires much effort. DOM must be diluted or degraded to a low concentration before the water is discharged. However, dilution requires a lot of water [40], and degradation requires maintaining aerobic conditions for a relatively long time. This means large amounts of air (oxygen) must be supplied to wastewater, and much energy and cost are necessary. Therefore, an alternative way to treat DOM in wastewater should be considered. Because it is well known that MiBs are able to adsorb and aggregate DOM onto their surfaces [4] in addition to supplying air (oxygen), we hypothesized that bubbling with MiBs would be an effective way to treat beverage-industry effluents, in which most of the organic matter is in the dissolved form. Relevant information is, nevertheless, still scarce.

In the present study, orange juice, lactic acid drink, and milk were selected as model DOM wastewaters from beverage industries, and degradation experiments were carried out. The decreases in DOM concentrations were measured during incubations of the three model substrates, and degradation patterns were compared with each other, because the three substrates containing different nitrogen contents showed quite different degradation patterns. Simultaneously, as it was revealed that some part of DOM was converted to particulate organic matter (POM) during the incubation under MiBs, amounts of POM both suspended in the water and floating on the water surface were determined. Finally, the relative effectiveness of MiBs versus MaBs for treating and purifying such wastewaters were also evaluated, and changes of bacterial abundances and several physicochemical factors were determined. Results of this study provide valuable information about MiBs for the effective DOM treatment of beverage-industry wastewater, in particular for the wastewater with high nitrogen contents.

For better and easier understanding, all the abbreviations used in the present study are listed in Table 1.

**Table 1.** List of the abbreviations used in the present study.

| | |
|---|---|
| ISO | International Organization for Standardization |
| MiB | microbubble |
| MaB | macrobubble |
| DO | dissolved oxygen |
| DOM | dissolved organic matter |
| POM | particulate organic matter |
| TOC | total organic carbon |
| DOC | dissolved organic carbon |
| POC | particulate organic carbon |
| S-POC | POC suspended in the water |
| F-POC | POC floating on the water surface |
| C/N ratio | carbon/nitrogen ratio |
| COD | chemical oxygen demand |

## 2. Materials and Methods

### 2.1. Experimental Apparatus

Two incubation containers (20 cm diameter, 100 cm height) were used for the experiments (Figure 1). Each container held 30 L of test solution which was circulated by a water pump, with the substrate liquid flowing in 15 cm above the bottom of the container and out 5 cm above the bottom.

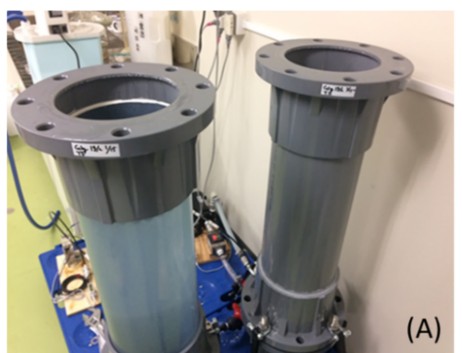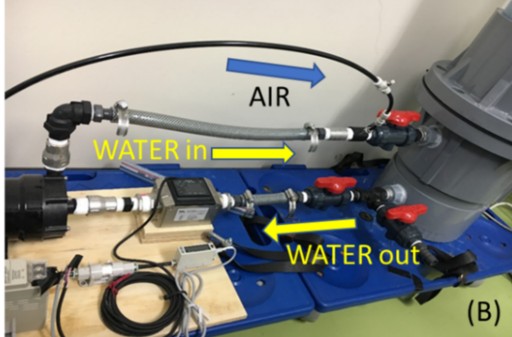

**Figure 1.** Photographs of the experimental apparatus. (**A**) A set of two incubation containers (30 L) was used. (**B**) Water was circulated by a water pump, and a microbubble generator was incorporated into the water circulation line of each container via the air line.

Microbubbles (MiBs) were generated via a gas–liquid shear system (FB-S15AI, Sakamoto-Giken Inc., Nankoku, Kochi, Japan) in which a gas shear field was created by swirling a liquid [41]. Output from the MiB generator was incorporated into the water circulation line of each container. Air supplied to the system was prefiltered through a 0.2 µm filter to remove microbial contamination from the atmosphere. The flow rates of air and water were 15 mL min$^{-1}$ and 20 L min$^{-1}$, respectively. Air MiBs with an average diameter of 50 µm were generated in this way and supplied to the systems (Figure 2). Details of the determination of the distribution of bubble sizes are as described by Mikasa et al. [41].

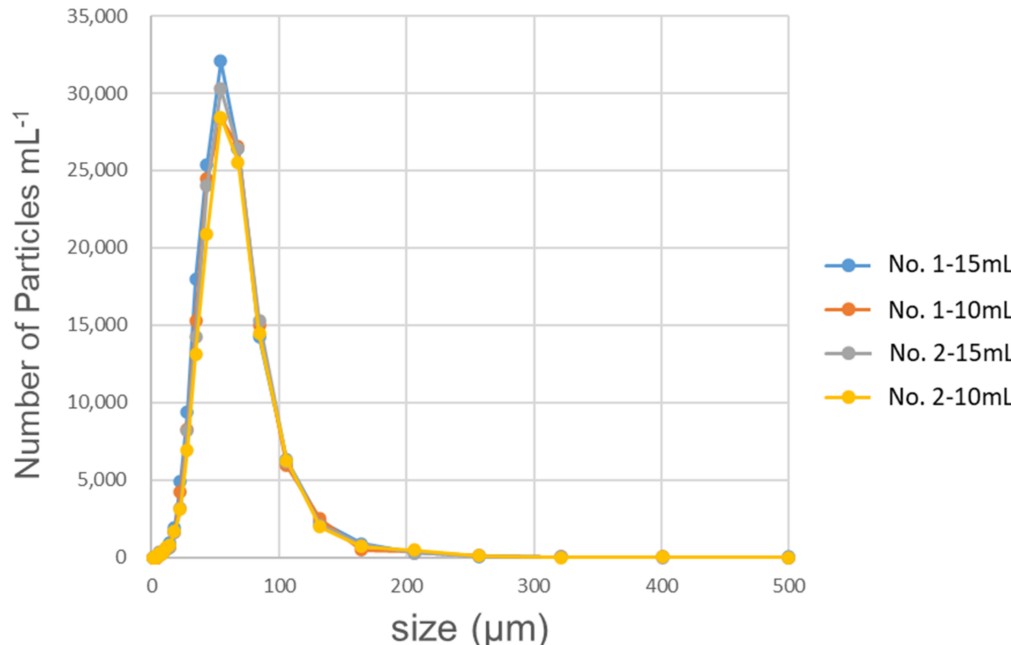

**Figure 2.** Frequency distribution of the diameter of microbubbles (MiBs) produced by two types of MiB generators. Water flow rates (WF) and air supply rates (AS) were as follows: "No. 1–15 mL": WF 20.5 L min$^{-1}$, AS 15 mL min$^{-1}$, "No. 1–10 mL": WF 20.5 L min$^{-1}$, AS 10 mL min$^{-1}$, "No. 2–15 mL": WF 19.5 L min$^{-1}$, AS 15 mL min$^{-1}$, "No. 2–10 mL": WF 19.5 L min$^{-1}$, AS 10 mL min$^{-1}$.

To compare the effect of MiBs versus macrobubbles (MaBs), MaBs were generated by an aquarium air pump (Super Silent Air Pump Pulse 4000, Kotobuki, Nara, Japan). The flow rates for water and air in the MaB containers were similar to those for the MiB experiments.

*2.2. Model Substrates and Incubations*

Commercially available orange juice (Ehime Beverage Inc., Ehime, Japan), lactic acid drink (Asahi Group Foods, Ltd., Tokyo, Japan), and milk (Snow Brand Milk Products Co., Ltd., Tokyo, Japan) were used as model wastewaters/substrates in this study. Table 2 shows the initial concentrations of organic carbon and nitrogen in the three substrates. Mean and SD values were obtained from six replicates. The organic carbon concentrations were 42.7 (SD ± 1.59), 43.0 (±1.56), and 63.6 (±1.99) g L$^{-1}$ in the orange juice, lactic acid drink, and milk, respectively. These concentrations of organic carbon were diluted for all experiments to ~0.5 g-C L$^{-1}$ using groundwater. The concentrations of organic carbon in the diluted effluents were therefore ~1% of the concentration in the original beverages. The total volume of water in each experimental container was set to 30 L for all substrates.

**Table 2.** Carbon and nitrogen content and C/N ratio of each beverage used in this research. Mean (±SD) values were obtained from six replicates. Different superscript letters in each column show significant differences ($p < 0.05$).

| | C (g-C L$^{-1}$) | | N (g-N L$^{-1}$) | | C/N |
|---|---|---|---|---|---|
| | **Mean** | **(±SD)** | **Mean** | **(±SD)** | |
| Orange Juice | 42.7 [a] | 1.59 | 1.34 [a] | 0.08 | 31.9 [a] |
| Lactic Acid Drink | 43.0 [a] | 1.56 | 0.67 [b] | 0.06 | 64.7 [b] |
| Milk | 63.6 [b] | 1.99 | 5.23 [c] | 0.17 | 12.2 [c] |

Each experiment was allowed to incubate for 10 days at room temperature (18–25 °C). Water temperature was not controlled during the experiments. Subsamples of test solutions

were collected at 0, 1, 3, 5, 7, and 10 days after starting the incubations. Some organic matter floated to the water surface with the bubbles during incubation. The floating materials were collected and removed from the container using a 100 mL conical beaker just before the subsamples were collected. Water subsamples (100 mL) were collected from the water surface using the conical beaker.

### 2.3. Determination of Organic Carbon

The organic carbon in the incubation containers was assigned to one of three categories: particulate organic carbon (POC) floating on the water surface (F-POC), POC suspended in the water (S-POC), and dissolved organic carbon (DOC).

The total amount of F-POC was determined by filtering all floating materials on GF/F glass fiber filters (47 mm diameter; Whatman, Maidstone, UK). After removal of floating materials, water samples were filtered through double-layered GF/F filters for determining S-POC. The amount of organic carbon in the lower filter paper (DOC) was subtracted from that of the upper filter (POC + DOC) as a blank value, because the amount of DOC absorbed to a filter paper was not negligible compared with the amount of POC retained on the filter paper. These filter papers were used for the determination of organic carbon with an NCH elemental analyzer (SUMIGRAPH NC-220F, Sumika Chemical Analysis Service. Ltd., Osaka, Japan). Concentrations of DOC in the filtrates were measured using a TOC analyzer (TOC-L CSH J 100, Shimadzu, Kyoto, Japan). Both S-POC and DOC were measured in duplicate or triplicate samples, and total amounts of S-POC and DOC were calculated by multiplying average concentrations by the total volume of water in the container. The total organic carbon (TOC: g-C) was calculated as the total sum of F-POC, S-POC, and DOC and was expressed as the amount of carbon in the container. All filters and filtrate samples were kept frozen ($-20\ °C$) until analysis.

In addition, subsamples (50 mL) of the water samples remaining after filtration for S-POC and DOC analyses were fixed with formalin (1% $v/v$) and refrigerated prior to determination of the bacterial abundances (Section 2.4).

### 2.4. Determination of Other Characteristics

Changes in bacterial abundances, DO concentration, and pH values were determined during the degradation experiments using the milk and lactic acid drink as two representative model substrates of high and low nitrogen contents. Bacterial abundances were determined in triplicate samples by a standard direct counting method, using the fluorescein dye DAPI (4′,6-diamidino-2-phenylindole; final concentration 1 µg mL$^{-1}$) (Sigma-Aldrich Japan, Tokyo, Japan) and an epifluorescence microscope (Olympus BX51-U-LH100HGAPO, Tokyo, Japan) [42]. The DO concentration and pH were determined on every sampling occasion, at a depth of 20 cm below the water surface using a DO meter (YSI-85, Xylem Japan, Kanagawa, Japan) and a pH meter (D-71, Horiba, Kyoto, Japan), respectively.

### 2.5. Statistical Significance

Differences of TOC concentrations and bacterial abundances between the containers bubbled with MiBs and MaBs were assessed for statistical significance by using Student's *t*-test. Differences were considered significant and very significant if the type I error rates (*p* value) were less than 0.05 and 0.01, respectively.

## 3. Results

Because there were only two containers and microbubble generators, replicated experiments could not be carried out simultaneously. The experiments were therefore performed two or three times in succession. However, as incubation conditions such as room temperature and time for collecting groundwater to dilute the substrates were different, which made it inappropriate to simply average the replication data, representative data are shown for the experiments.

### 3.1. Changes in the Amount of Organic Carbon

Figure 3 shows changes in the amounts of F-POC, S-POC, and DOC during the 10-day incubations for each of the three substrates. In the orange juice and lactic acid drink, most of the organic carbon was initially in the dissolved form, and particulate materials were scarce (Figure 3A,B). In contrast, a relatively large amount of S-POC was detected in the milk (Figure 3C).

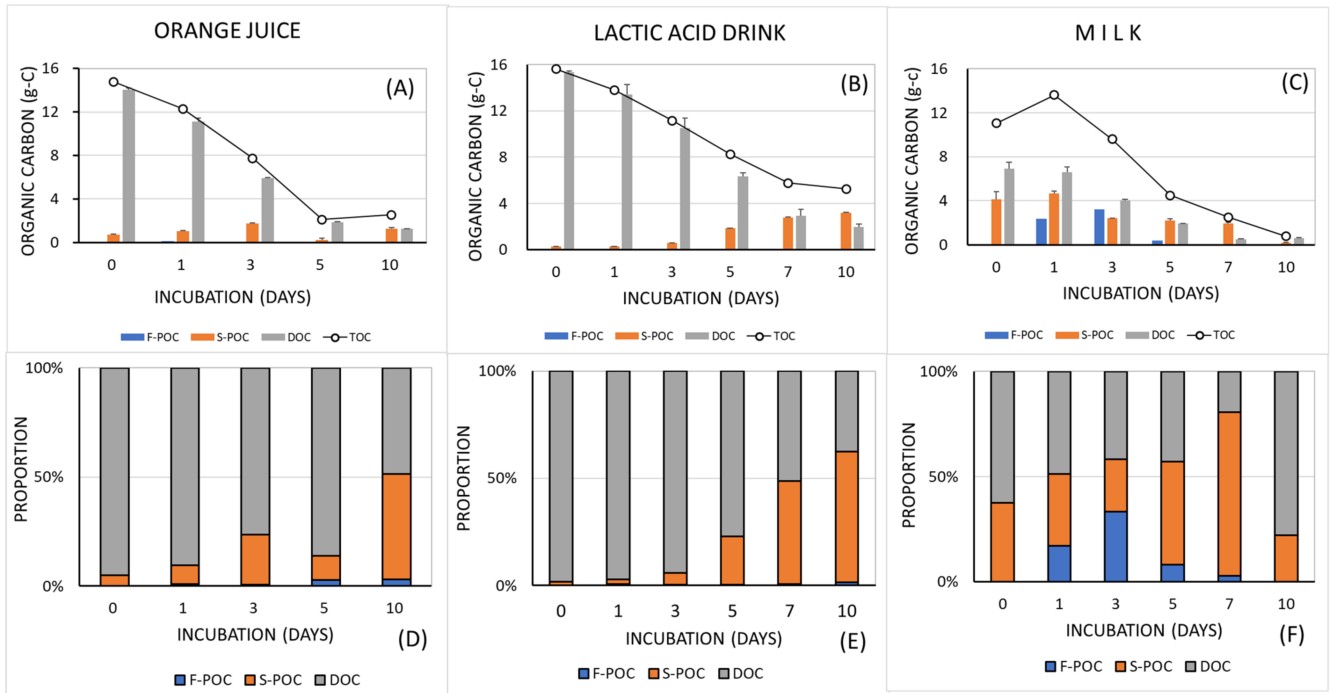

**Figure 3.** Changes in the amounts of TOC, F-POC, S-POC, and DOC during incubation of three beverage substrates with microbubble treatment (**A–C**), and proportional distribution of the three types of organic carbon materials (**D–F**).

During the incubation period, the total amount of DOC in the containers decreased rapidly, from 14.0 (day 0) to 1.24 g-C (day 10) in the orange juice (Figure 3A), from 15.3 to 1.98 g-C in the lactic acid drink (Figure 3B), and from 6.9 to 0.59 g-C in the milk (Figure 3C). The amount of S-POC increased during incubation in the orange juice and lactic acid drink but decreased in the milk. An interesting result was the increased F-POC content during days 1 to 5 in the milk (Figure 3C).

Initially, more than 90% of the organic carbon was in the form of DOC in the orange juice and lactic acid drink (Figure 3D,E). The percentages of DOC decreased gradually to 48.6% (orange juice) and 37.6% (lactic acid drink) by the tenth day. However, the division of organic carbon types in the milk followed a different temporal pattern. Unlike the orange juice and the lactic acid drink, in which the proportions of particulate materials (F-POC and S-POC) were relatively large at the end of the incubation period (Figure 3D,E), in milk substrate, the proportion of F-POC and S-POC dropped sharply on day 10 (Figure 3F).

Figure 4 shows the cumulative amounts of F-POC during the 10-day incubations. Although very little F-POC was detected in the orange juice and lactic acid drink, a dramatic increase of F-POC occurred in the milk during the first three days, but there was no subsequent increase. During the 10-day incubation period, the TOC (F-POC + S-POC + DOC) decreased to 17.3% (orange juice), 33.7% (lactic acid drink), and 6.9% (milk) of the initial TOC values (Figure 3A–C).

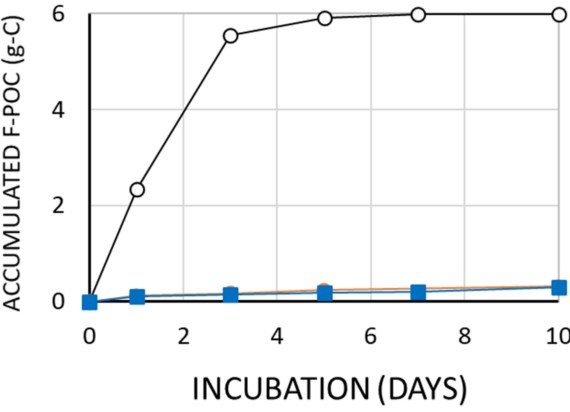

**Figure 4.** Changes in the accumulated quantity of F-POC in three tested liquids during 10 days of incubation.

### 3.2. Effects of Microbubbles vs. Macrobubbles

As mentioned in Section 3.1, the degradation patterns in the lactic acid drink and the milk were quite different (Figure 3B,E and Figure 3C,F, respectively). There were large differences in the concentrations of organic carbon and nitrogen as well as in the C/N ratios (Table 2). We therefore carried out degradation experiments using the two substrates (lactic acid drink and milk) and compared the effects of MiBs versus MaBs.

Figure 5 shows the changes in the amounts of TOC and accumulated F-POC in the MiB and MaB treatments. In the lactic acid drink, which contained little organic nitrogen, the differences between the MiB and MaB treatments were small (Figure 5A,B). However, differences between the MiB and MaB treatments were clearly apparent in the milk, which has a high nitrogen content (Figure 5C,D). During the 10-day incubation, the TOC decreased by ~93.1% in the MiB treatment, whereas the TOC decreased by only 37.0% in the MaB treatment (Figure 5C). Differences between the two treatments were significant ($p < 0.05$) on the third and fifth days, and very significant ($p < 0.01$) on the seventh and tenth days. The accumulation of F-POC was also remarkably different between the two treatments; there was very little production of F-POC in the MaB treatment compared with the MiB treatment (Figure 5D).

Fluctuations of DO concentrations, abundances of bacteria, and pH values are illustrated in Figure 6. In both the lactic acid drink and the milk, DO concentrations decreased drastically in the first 24 h and remained at nearly zero throughout the experiment in the MaB treatment, whereas the DO started to increase again after the fifth or seventh day in the MiB treatment. There was a significant difference ($p < 0.01$) in bacterial abundance between the two treatments on most days in both liquid substrates (Figure 6B,E). Bacterial numbers in the MiB treatment were more than an order of magnitude greater than those in the MaB treatment up to day 5 inclusive, as the TOC in MiB treatment still remained higher (~5 g-C) during this period (Figure 5C). However, bacterial abundances decreased along with the decrease of TOC in MiB treatment. Values of pH also differed between the MiB and MaB treatments (Figure 6C,F). In both treatments, the pH values decreased rapidly during the first three days. Thereafter, the pH in the MiB treatment was higher than the MaB in both substrates, remaining near 5.5 to 6.0 in the lactic acid drink and rising to near 7 in the milk (Figure 6C,F).

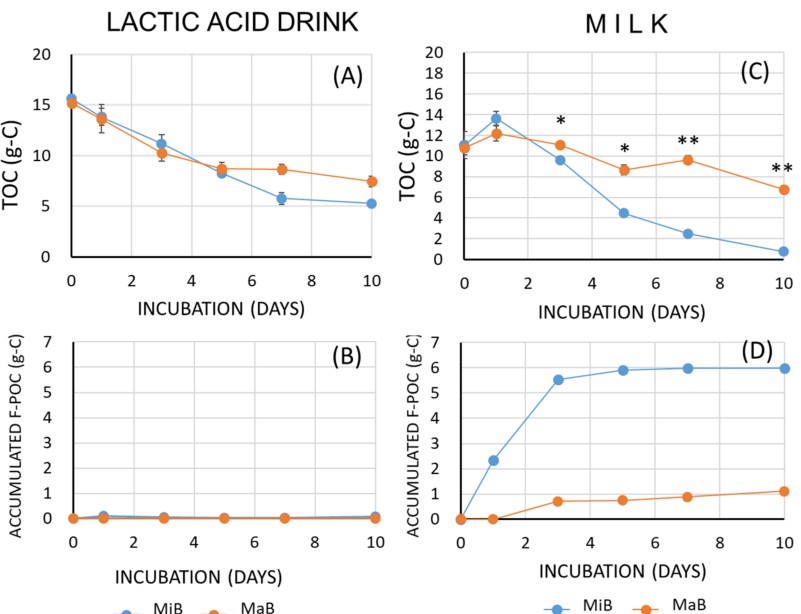

**Figure 5.** Changes in the amount of TOC (**A**,**C**) and the accumulated quantity of F-POC (**B**,**D**) under MiB or MaB treatments of two substrates during 10 days of incubation. Statistical significances are shown at $p < 0.05$ (*) or $p < 0.01$ (**) levels.

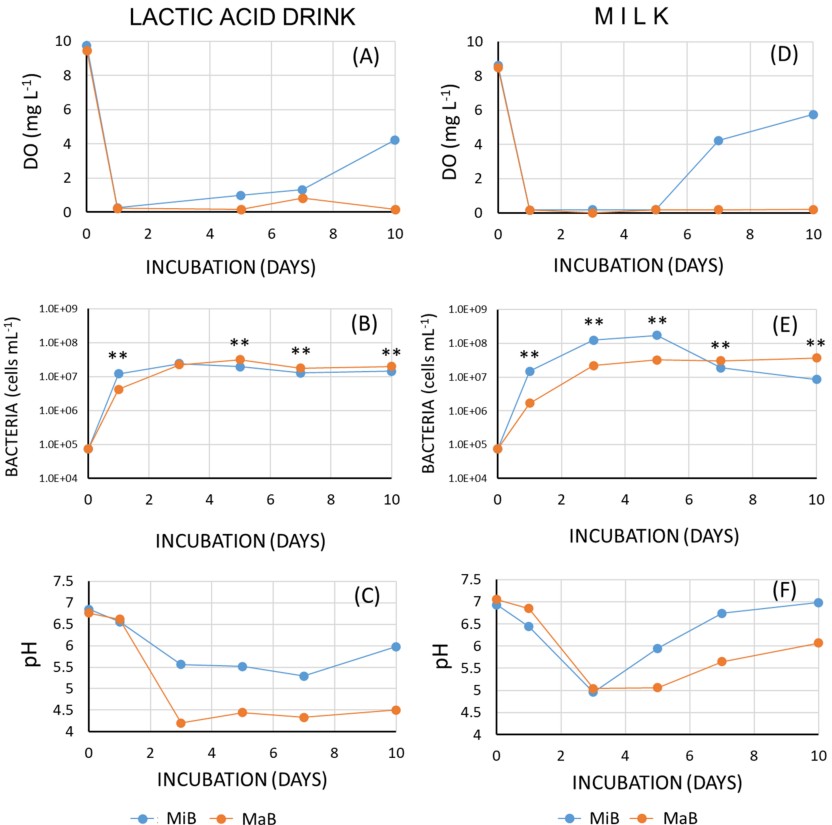

**Figure 6.** Changes in dissolved oxygen (DO) (**A**,**D**), bacterial abundances (**B**,**E**), and pH values (**C**,**F**) in the MiB and MaB treatments of two substrates during 10 days of incubation. Statistical significances are shown at $p < 0.01$ (**) levels.

## 4. Discussion

In the present study, three model substrates—orange juice, lactic acid drink, and milk—were selected, and the degradation of these substrates was investigated to evaluate the effectiveness of MiBs for the treatment and purification of wastewater from beverage industries. Results indicated that the amount of TOC was lowest in the milk after a 10-day incubation. Only 0.76 g-C (6.9%) remained, i.e., 10.3 g-C (93.1%) of the initial TOC had been removed (Figure 3C). In contrast, 5.26 g-C (33.7%) of organic carbon still remained in the lactic acid drink after the 10-day incubation (Figure 3B). The amount of residual organic carbon remaining in the orange juice was intermediate between these two (2.55 g-C, or 17.3% (Figure 3A)). These results suggest that organic matter with a relatively low C/N ratio (with a high organic nitrogen content) was degraded more quickly in wastewater than organic matter with a relatively high C/N ratio (with a low organic nitrogen content) (Table 2). This result is supported by the fact that bacterial abundances were more than $10^8$ cells mL$^{-1}$, which were an order of magnitude greater in the milk than in the lactic acid drink on the third and fifth days of the incubations in the MiB treatment (Figure 6B,E). These results suggest that wastewater containing high concentrations of proteins and amino acids may be purified more easily than wastewater containing mainly carbohydrates.

The results of the MiB and MaB treatments on two representative substrates, lactic acid drink and milk, indicate that the decrease of TOC occurred much faster in the MiB treatment, and bacterial growth was more rapid and bacterial abundance higher in the MiB treatment than the MaB treatment (Figures 5 and 6). During the incubation period, the concentration of DO decreased to zero, and the pH also decreased rapidly (Figure 6). The decrease in pH may have been due to the production of organic acids by anaerobic decomposition (fermentation) in the absence of oxygen. However, whereas the MaB treatment remained anaerobic, and the pH remained relatively low, the MiB treatment returned to aerobic conditions quickly, and pH values rose to 6.0–7.0 by the end of the incubation period (Figure 6D,F). These results suggest that MiBs did a much better job than MaBs of supplying oxygen and accelerating the growth of aerobic bacteria when air was bubbled at the same rate (15 mL min$^{-1}$).

The difference between the MiB and MaB treatments was larger in the milk than in the lactic acid drink (Figures 5 and 6). The explanation for this difference may be that wastewater with nitrogen-rich compounds (milk) supported more rapid bacterial growth than wastewater with nitrogen-poor compounds (lactic acid drink), and, consequently, the former accelerated the degradation of the organic matter in wastewater.

Another interesting point is the production of F-POC. Much F-POC was detected in the container with the milk substrate, but there was very little F-POC in the orange juice, and almost no F-POC in the lactic acid drink (Figures 3 and 4). A similar result producing high amounts of F-POC was obtained in the peptone solution, too (data not shown). One strategy for removing suspended solids (SS) from wastewater that contains high concentrations of SS is to separate the SS by adsorption and filtration using membranes [35,38]. However, filtration is associated with several problems, such as membrane fouling due to clogging by deposited materials [37,43]. Previous studies have reported the use of nano- and microbubbles to separate small particles of palm oil [8] and minerals [39,44] on the surface of fine bubbles and remove them by floatation from waste effluents. In addition, the surfaces of MiBs have a tendency to adsorb, condense, and aggregate dissolved organic materials [38]. Results of the present study (Figures 3 and 4) suggest that aggregating DOM on bubble surfaces and separating it by floatation may be an effective way to remove DOM from wastewater with a high nitrogen content.

The standards for wastewater drainage promulgated by the Ministry of the Environment, Japanese Government, indicate that effluent from industries must contain no more than 160 mg L$^{-1}$ of chemical oxygen demand (COD) [45]. This standard corresponds to 60 mg L$^{-1}$ of C as TOC. In the present study, the amounts of TOC remaining after a 10-day incubation were 0.76 g-C in the milk, which corresponded to 27 mg-C L$^{-1}$ of TOC, 2.55 g-C (95 mg-C L$^{-1}$ of TOC) in the orange juice, and 5.26 g-C (195 mg-C L$^{-1}$ of

TOC) in the lactic acid drink (Figure 3). These results indicate that only the milk-based research substrate after MiB treatment would have passed the Japanese legal standard. However, the initial concentration of organic carbon in this study was reduced to around 0.5 g-C L$^{-1}$, which was about 1% of the initial concentration in the original beverages. These initial TOC concentrations may have been much higher than that of actual industrial effluent. Moreover, we supplied air at 15 mL min$^{-1}$ to 30 L of water, which is equivalent to only 0.05% (*v/v*) of air supplied to the wastewater per minute. If the concentrations of organic carbon in the actual effluents discharged by beverage industries are less than half or one-third of the initial concentrations of the present study, and effluents are treated with larger volumes of air (more than 0.05% min$^{-1}$), a much more rapid decrease and a lower amount of the remaining TOC will be surely achieved to a level below the Japanese legal standard.

When we conceive the practical treatment facility of beverage wastewater of 300 m$^3$ ($10^4$ times greater than the present study), usual MaBs aeration equipment will cost about USD 62,000, while that with MiBs costs about USD 66,400 [46]. This means that only 7% cost difference and no significant increase in the expense would be borne by beverage industries. Therefore, when DOM from beverage wastewater is treated and purified by similar costs, aeration by MiBs is much more effective than that by usual MaBs.

## 5. Conclusions

Results of the present study indicate that treatment with microbubbles—roughly a few tens of micrometers in diameter—is much superior to that with macrobubbles in supplying oxygen, accelerating the growth of aerobic bacteria, and consequent degradation of DOM. It is also suggested that DOM containing more nitrogenous compounds was purified more effectively than that with less nitrogen by microbial degradation and floating separation. Microbubble treatment, therefore, is a promising method to stimulate the decomposition of DOM in the wastewater from beverage industries, which often contains a very high content of organic carbon. The treatment process may successfully reduce the COD to the legally allowed level for discharge to the natural environment.

**Author Contributions:** Conceptualization, K.F.; methodology, K.F., T.O., K.M. and T.M.; validation, K.F.; formal analysis, K.F.; investigation, K.F. and T.H.; resources, T.M. and M.S.; data curation, K.F. and T.O.; writing—original draft preparation, K.F.; writing—review and editing, K.F.; visualization, K.F.; supervision, T.H.; project administration, K.F. and T.H.; funding acquisition, T.H. and M.S. All authors have read and agreed to the published version of the manuscript.

**Funding:** This study was conducted with support from the Kochi Research and Development Fund, and part of this study was financially supported by the Four-Dimensional Kuroshio Marine Science (4D-KMS) research project, Kochi University, and the "research fund for the director", the Open University of Japan.

**Institutional Review Board Statement:** Not applicable.

**Informed Consent Statement:** Not applicable.

**Acknowledgments:** The authors thank Kochi Prefecture, Japan, for their financial support of this study. The authors also give their sincere gratitude to the Four-Dimensional Kuroshio Marine Science (4D-KMS) research project, Kochi University, and the Open University of Japan, for their financial support. The authors thank Kenji Yamamoto and Iwao Mitani, Sakamoto Giken Inc., for their technical support of microbubble-generating apparatus and valuable discussion, too.

**Conflicts of Interest:** The authors declare no conflict of interest. The funders had no role in the design of the study; in the collection, analyses, or interpretation of data; in the writing of the manuscript, or in the decision to publish the results.

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
