# Peer review of "Effective Purification of Eutrophic Wastewater from the Beverage Industry by Microbubbles"

_water, doi:10.3390/w13243661_

Round 1

Reviewer 1 Report

Reviewer 1:

I recommend major amendments at this level.

General comments:

I reviewed the manuscript entitled “Effective Purification of Eutrophic Wastewater from the  Beverage Industry by Microbubbles”. The work carried out in the manuscript is interesting and aimed at using orange juice, lactic acid drink, and milk to prepare model wastewaters from beverage industries, and carried out degradation experiments using MiBs. However, there are several remarks that authors should follow before possible publication. The manuscript needs a thorough revision of its language and style. One of the main problems of the manuscript is the lack of sufficient and relevant references, especially in the discussion section. The manuscript has a lot of information however there are some lacking connectors. I suggest authors take a closer look and adjust the write-up to be more precise and appealing to the readers. Additionally, the novelty of the research still is not clear and the discussion and conclusions cannot satisfy me. This shows the manuscript lacks the new and interesting insights to deliver.  It is better to do not to use the first person's pronoun. Do not use "we, us, or our" throughout the paper. Please also remove ANY lumped references. Please define each of them separately to avoid inappropriate citations. Too many abbreviations are used. I recommend a nomenclature section for the abbreviations and variables used throughout the passage. Highlights are necessary for this journal. Please provide a graphical abstract to provide a visual summary of the main findings of the study. The journal's author guidelines and instructions should be followed in preparing the revised version. Some other issues that need to be addressed are:

Detailed comments:

Title: Ok.

Abstract:

Please improve the abstract. The abstract should have one sentence per each: context and background, motivation, hypothesis, methods, results, conclusions. In the abstract, please add an indication of the achievements from your study that are relevant to the journal scope. Please be concise - maximum 1-2 lines. Please explain the contributions of the study in the abstract.

Introduction:

The introduction is not well presented and also not strongly linked to the gaps of the research, therefore the novelty of the work is not significant. Please improve the state of the art overview, to clearly show the progress beyond the state of the art. The lack of proper justification creates the wrong impression that the authors are unaware of the recent developments. A high-quality paper has to provide a proper state-of-the-art analysis after the literature review and only based on the analysis to formulate the paper goals. In addition; the introduction should be clearly stated research questions and targets first. Then answer several questions: Why is the topic important (or why do you study on it)? What are research questions? What has been studied? What are your contributions? The major defect of this study is the debate or argument is not clearly stated in the introduction session. At the end of the introduction, the statement of the paper’s goal and the explanation of novelty has to be properly formulated. Currently, this is not performed well. The aim of the introduction should improve.

I would suggest to the authors, please provide one table and compare your current results with others.

Could you please clarify the cost of this method? Is it economy? Does can be applied on a large scale? How?

Materials and Methods:

Please avoid having one heading after another with no discussion in between as in the case of Section 2 and 2.1. Kindly inspect the entire document for similar instances and revise accordingly. Please add in the beginning your scientific hypothesis. In the course of describing the performed actions, please provide reader guidance, sufficient for understanding why those actions have been performed.

Results and discussion:

The structure of this work should be reorganized. For example, a Section of results should be combined with Discussion. The authors are suggested to have the results and discussion part together. This section is the second main problem of the current manuscript. These are only some general comments that might help the authors to improve in the next cycle. All the obtained results need to discuss along with the findings of other researchers. Overall the authors generated enough data. However, the presentation of those is not scientifically strong. The entire Discussions section is generally weak and must be strengthened by discussing further and citing more best practices from the literature.

Conclusions:

Where is the conclusion and future works?? Please provide separately with a proper heading in the next revision.

References:

Bibliography style is not always consistent, please check the reference section carefully and correct the inconsistency.

Author Response

Reviewer 1:

I recommend major amendments at this level.

General comments:

I reviewed the manuscript entitled “Effective Purification of Eutrophic Wastewater from the Beverage Industry by Microbubbles”. The work carried out in the manuscript is interesting and aimed at using orange juice, lactic acid drink, and milk to prepare model wastewaters from beverage industries, and carried out degradation experiments using MiBs. However, there are several remarks that authors should follow before possible publication. The manuscript needs a thorough revision of its language and style. One of the main problems of the manuscript is the lack of sufficient and relevant references, especially in the discussion section. The manuscript has a lot of information however there are some lacking connectors. I suggest authors take a closer look and adjust the write-up to be more precise and appealing to the readers. Additionally, the novelty of the research still is not clear and the discussion and conclusions cannot satisfy me. This shows the manuscript lacks the new and interesting insights to deliver.  It is better to do not to use the first person's pronoun. Do not use "we, us, or our" throughout the paper. Please also remove ANY lumped references. Please define each of them separately to avoid inappropriate citations. Too many abbreviations are used. I recommend a nomenclature section for the abbreviations and variables used throughout the passage. Highlights are necessary for this journal. Please provide a graphical abstract to provide a visual summary of the main findings of the study. The journal's author guidelines and instructions should be followed in preparing the revised version. Some other issues that need to be addressed are:

     Thank you for your valuable comments and suggestions. The authors revised MS according to your suggestions as follows;

  1. For showing clearly the purpose and the novelty of the research, abstract was changed almost completely, and some explanations were added to the Introduction (line 72-95). In addition, some parts of the Discussion were also revised as relating to and emphasizing the novelty described in the Introduction.
  2. [6. Conclusion] was added.
  3. The active voice (i.e., we did something) was replaced by the passive voice style throughout the text.
  4. Although the Reviewer 1 commented “One of the main problems of the manuscript is the lack of sufficient and relevant references, especially in the discussion section.”, we suppose that sufficient and relevant references were cited in the Introduction. For the 1st, the 2nd, and the 3rd paragraphs of the Discussion, because no appropriate references on the decrease of DOC relating to nitrogen contents and changes in DO and pH during the decomposition of organic matters are available, we did not give any references. However, in the 4th paragraph on the floatation of SS and POM produced by DOM adsorption and aggregation on the bubble surface and floatation on the water surface, several important references were cited there. Please let me know any appropriate references related to the first 3 paragraphs of Discussion if you are not yet satisfied with.
  5. To the comment of “Please also remove ANY lumped references.”, we will answer that only one part cited 4 references (l. 73) and 3 parts gave 3 references (l. 41, 44 and 49). Therefore, we do not think there are many “lumped references”.
  6. For better and easy understanding of abbreviations, Table 1 was newly added.
  7. We did not find any direction that highlights are necessary in the “Instructions for Authors” for the journal of “Water”. But 5 highlights were prepared according to your suggestion.
  8. A graphical abstract was also newly prepared.
  9. We had submitted this article to another journal at first. However, the editorial board of that journal recommended “Water” and transferred it to the editorial board of “Water”. Therefore, some part of the article style was not fitted to “Water”. We arranged it to the style of the journal.

Detailed comments:

Title: Ok.

Abstract:

Please improve the abstract. The abstract should have one sentence per each: context and background, motivation, hypothesis, methods, results, conclusions. In the abstract, please add an indication of the achievements from your study that are relevant to the journal scope. Please be concise - maximum 1-2 lines. Please explain the contributions of the study in the abstract.

Because the “Instructions for Authors” directs that the abstract should be a total of about 200 words maximum, we had obeyed it. However, almost all articles newly published in Water gave the abstract with more than 200 words and the maximum numbers was 355 words! Therefore, more detailed abstract including your demands was prepared. We hope this abstract will be accepted by the Editorial Board.

Introduction:

The introduction is not well presented and also not strongly linked to the gaps of the research, therefore the novelty of the work is not significant. Please improve the state of the art overview, to clearly show the progress beyond the state of the art. The lack of proper justification creates the wrong impression that the authors are unaware of the recent developments. A high-quality paper has to provide a proper state-of-the-art analysis after the literature review and only based on the analysis to formulate the paper goals. In addition; the introduction should be clearly stated research questions and targets first. Then answer several questions: Why is the topic important (or why do you study on it)? What are research questions? What has been studied? What are your contributions? The major defect of this study is the debate or argument is not clearly stated in the introduction session. At the end of the introduction, the statement of the paper’s goal and the explanation of novelty has to be properly formulated. Currently, this is not performed well. The aim of the introduction should improve.

Introduction was revised so as that all your demands described as above were included.

The purpose of the present study was shown more clearly (l. 72-97). That is, treatment of DOM by usual aeration (MaBs) needs much effort and cost but treatment using MiBs will be more effective when the effort and cost are similar. In addition, the difference of the patterns of DOM degradation depends on its nitrogen contents.      

I would suggest to the authors, please provide one table and compare your current results with others.

Unfortunately, no current data comparable to the present study were available. Therefore, we are afraid such a table will not be convenient.

Could you please clarify the cost of this method? Is it economy? Does can be applied on a large scale? How?

Comparison of the cost of wastewater treatment using microbubbles and usual macrobubbles aerations was added in the last part of Discussion (l. 332-337).

Materials and Methods:

Please avoid having one heading after another with no discussion in between as in the case of Section 2 and 2.1. Kindly inspect the entire document for similar instances and revise accordingly. Please add in the beginning your scientific hypothesis. In the course of describing the performed actions, please provide reader guidance, sufficient for understanding why those actions have been performed.

I am sorry I do not understand your meaning. We are sure that each section (2.1.-2.5.) of Materials and Methods describes enough information for them.

Results and discussion:

The structure of this work should be reorganized. For example, a Section of results should be combined with Discussion. The authors are suggested to have the results and discussion part together. This section is the second main problem of the current manuscript. These are only some general comments that might help the authors to improve in the next cycle. All the obtained results need to discuss along with the findings of other researchers. Overall the authors generated enough data. However, the presentation of those is not scientifically strong. The entire Discussions section is generally weak and must be strengthened by discussing further and citing more best practices from the literature.

Judging from our intention, we suppose that descriptions of Results and Discussion should be separated. For the literature cited, please see our No. 4 comment for your General comments.

Conclusions:

Where is the conclusion and future works?? Please provide separately with a proper heading in the next revision.

Conclusion was added at the last part.

References:

Bibliography style is not always consistent, please check the reference section carefully and correct the inconsistency.

Reference list was checked again and some parts were revised.

Reviewer 2 Report

After paper review, there were several comments appeared:

  1. The aeration in biological reactors (if removal of carbon or nutrients is considered) are traditionally applied to a mixed liquor of wastewater and active sludge, but not only a wastewater. This approach is applied for food industry as well. Only aeration of wastewater seems to me not enough for efficient treatment.
  2. It is not a novel idea that fine bubbles provide better (than transport of oxygen. At list for biological reactors.
  3. According to above-mentioned, I suppose it is necessary to extend paper with information on entire treatment process, but not only aeration

Author Response

Reviewer 2:

After paper review, there were several comments appeared:

     Thank you for your valuable comments and suggestions. The authors revised MS according to your suggestions as follows;

  1. The aeration in biological reactors (if removal of carbon or nutrients is considered) are traditionally applied to a mixed liquor of wastewater and active sludge, but not only a wastewater. This approach is applied for food industry as well. Only aeration of wastewater seems to me not enough for efficient treatment.

     As described in the Introduction (l. 72-84), the purpose of the present study is to treat and purify the dissolve organic matter among various components of wastewater. Because sludge and other particulate matters are relatively easy to remove, but dissolve matters are often difficult to treat and some special techniques are necessary. And main part of wastewater from beverage industries is DOM. Therefore, we focused on DOM treatment here.  

  1. It is not a novel idea that fine bubbles provide better (than transport of oxygen. At list for biological reactors.

     Roles of microbubbles are not only to provide air (oxygen) but also to produce particulate matters by absorbing, condensing, and aggregating of DOM, and floating them to the water surface. This second role is more or less additional but important effect of microbubbles. We found that this second effect was more evident for the wastewater with high nitrogen contents. This is one of the novel findings in our study.

  1. According to above-mentioned, I suppose it is necessary to extend paper with information on entire treatment process, but not only aeration.

     Same answer as to the 1st and 2nd comments.

Reviewer 3 Report

Comment 1#: The substrate used as model wastewater/substrate are final products. However, the wastewater of beverages may contain residual fats, cleaners, solvents, preservatives, additives, color, flavors, etc. which are responsible for the high COD in wastewater. Consider it for line number 307-308.

Comment 2#: Rewrite the caption of figures 2, 5.

Comment 3#: Conclusion is an important to segment for the research article. Make it separate.

Comment 4#: More typo errors in the manuscript. Check it very carefully.

Comment 5#: The article after making the suggested changes is acceptable for publication.  

Author Response

Reviewer 3:

Thank you for your valuable comments and suggestions. The authors revised MS according to your suggestions as follows;

Comment 1#: The substrate used as model wastewater/substrate are final products. However, the wastewater of beverages may contain residual fats, cleaners, solvents, preservatives, additives, color, flavors, etc. which are responsible for the high COD in wastewater. Consider it for line number 307-308.

     Main effects of microbubbles are to supply oxygen (air), to accelerate the growth of aerobic bacteria and consequent degradation organic matters. Therefore, if such solvents, preservatives, additives, artificial color, etc. are refractory, even microbubbles have few (or no) effects. Please understand that we treated labile but large amounts of DOM, which mainly comprises beverage wastewater, in our study.    

Comment 2#: Rewrite the caption of figures 2, 5.

     Captions of figures 2 and 5 were revised.

Comment 3#: Conclusion is an important to segment for the research article. Make it separate.

Conclusion was added at the last part.

Comment 4#: More typo errors in the manuscript. Check it very carefully.

     I have checked the whole text carefully again and revised the wrong parts.

Comment 5#: The article after making the suggested changes is acceptable for publication.  

     Thank you for your valuable comments and suggestions again.

Reviewer 4 Report

Article entitled Effective Purification of Eutrophic Wastewater from the Beverage Industry by Microbubbles written by Kimio Fukami, Tatsuro Oogi, Kohtaro Motomura, Tomoka Morita, Masaoki Sakamoto and Takashi Hata and submitted to Water journal as a draft no 1487319 deals with an important issue of effective industrial wastewater treatment.

As English is not my native language, I am not able to assess language correctness.

The article is interesting and could be considered for publication in Water journal. However, while reading, I found some statements missing, confusing or unclear. Below I enclose the list of my comments. 

The title of the article suggests that real wastewater from the beverage industry was used. The Authors, however, in the abstract and the content of the article clearly indicate that it is a simulated wastewater. I suggest to include this in the title of the article.

Fig. 2. “old” and “new” it is not clear, more detailed description is needed.

Table 1. what is a, b and c?

The time required for the purification is long, the Authors report a duration of 10 days. Does the proposed treatment have application potential? It seems to me that it is worth discussing the results in comparison to other currently used treatment technologies.

How do the Authors explain the increase in carbon content in the case of the milk day 1-2 experiment (fig.3c) and lactic acid experiment day 5-10 (fig 3b)?

What is the source of the increase in suspensions of figs 3d, days 5-10? Similarly, for 3f.

What is the TOC removal mechanism according to the Authors? Is it about the oxidation itself or the mechanism involves biodegradation with the use of specific microorganisms. It seems likely that they are microorganisms. Why were detailed microbiological tests not performed? This must absolutely be completed. The Authors write about it in chapter 2.4, but I did not find the results (excluding figure 6b, d) or their discussions in the content of the article, and I consider it crucial. I believe that the experiment should be repeated with increased control of the microbiology of the process. You should even focus on it.

I don't see the results related to the nitrogen content.

Many of the statements contained in the discussion, although they seem correct, are pure speculation from the point of view of potential phenomena occurring during treatment - the Authors did not control the key parameters of the process, as well as what specific organisms developed.

This article is missing conclusions - I haven't found any conclusion chapter.

The purpose of the research is not clear to me. It is well known that the smaller the bubbles, the better the air / oxygen mass transfer to the wastewater and it is crucial for biological treatment. The scientific novelty must be clearly marked in the article.

Based on my comments and overall impression, I suggest major revision.

Author Response

Reviewer 4:

Article entitled Effective Purification of Eutrophic Wastewater from the Beverage Industry by Microbubbles written by Kimio Fukami, Tatsuro Oogi, Kohtaro Motomura, Tomoka Morita, Masaoki Sakamoto and Takashi Hata and submitted to Water journal as a draft no 1487319 deals with an important issue of effective industrial wastewater treatment.

As English is not my native language, I am not able to assess language correctness.

The article is interesting and could be considered for publication in Water journal. However, while reading, I found some statements missing, confusing or unclear. Below I enclose the list of my comments. 

     Thank you for your valuable comments and suggestions. The authors revised MS according to your suggestions as follows;

The title of the article suggests that real wastewater from the beverage industry was used. The Authors, however, in the abstract and the content of the article clearly indicate that it is a simulated wastewater. I suggest to include this in the title of the article.

     I think the title is the briefest abstract of the study and it includes the final goal of the study, too. When the title is changed to add “using model substrates” at the end of title, I am afraid it looks like strange. Moreover, abstract and Introduction have been already revised to show clearly the purpose of the present study, that is to treat and purify the dissolve organic matter among various components of wastewater (l. 14-34, 72-87). Please understand we leave the title as it.

Fig. 2. “old” and “new” it is not clear, more detailed description is needed.

     Expressions of “old” and “new” were changed to “No. 1” and “No. 2”.

Table 1. what is a, b and c?

     They show significant differences. Please see the last part of the table caption. [Different superscript letters in each column show significant differences (p<0.05).]

The time required for the purification is long, the Authors report a duration of 10 days. Does the proposed treatment have application potential? It seems to me that it is worth discussing the results in comparison to other currently used treatment technologies.

     Experimental conditions in the present study were too severe compared with the practical conditions of wastewater treatment. The initial concentrations of organic carbon (0.5 g-C L-1) were as much as 1 % of the original beverage and they were too high, and air supplying rate was only 0.05 % (v/v) min-1 of treating wastewater. We are sure that TOC will decrease more quickly by treating wastewater of less DOM contents and increasing air supply rates. Some explanation was added in the Discussion (l. 324-331).

How do the Authors explain the increase in carbon content in the case of the milk day 1-2 experiment (fig.3c) and lactic acid experiment day 5-10 (fig 3b)?

     For the TOC increase in milk (Figure 3C), we suppose the value of the first day of 11.06 g-C was accidentally low because the initial concentrations of three substrates were adjust to around 15 g-C (0.5 g-C L-1 x 30 L) and it should be higher than 11.06 g-C. Therefore, it may not be “increase”. Increase from the 5th to the 10th days in orange juice (Figure 3A) (not lactic acid drink, Figure 3B) may be within the error range.

What is the source of the increase in suspensions of figs 3d, days 5-10? Similarly, for 3f.

     These values are not the actual concentrations but the % proportion. Therefore, it is OK that the relative values fluctuate day by day. The reason why the proportions of S-POC increased during the incubation would be that particulate organic matter was newly produced by absorbing, condensing, and aggregating of DOM, and the growth of bacteria (bacterial cells were retained on filter papers and measured as S-POC).

What is the TOC removal mechanism according to the Authors? Is it about the oxidation itself or the mechanism involves biodegradation with the use of specific microorganisms. It seems likely that they are microorganisms. Why were detailed microbiological tests not performed? This must absolutely be completed. The Authors write about it in chapter 2.4, but I did not find the results (excluding figure 6b, d) or their discussions in the content of the article, and I consider it crucial. I believe that the experiment should be repeated with increased control of the microbiology of the process. You should even focus on it.

     As you commented, there are two mechanisms for TOC removal; one is degradation by aerobic bacteria and the other is floating separation, because all the F-POC were removed at every sampling occasion (see 2.4, l. 146-148, 155-156). However, detection of F-POC was only in the milk and almost no F-POC were detected in the lactic acid drink and orange juice. Therefore, main cause of TOC decrease will be degradation by aerobic bacteria. We did not use or add “special microorganisms” in the experiments. Source of all bacteria was groundwater which we used for dilution of model beverage substrates. During the incubation, many aerobic bacteria grew and they decomposed organic matter. However, the qualitative information on bacteria (community structure) does not matter and abundance is important for our purpose. Therefore, we determined the fluctuation of bacterial abundances in the experiments using milk and lactic acid drink as representative beverages with different nitrogen contents (see 2.4., l. 174-176).  

I don't see the results related to the nitrogen content.

     We focused on the amount of organic matter and followed the decrease in TOC concentration. If we show and discuss on the nitrogen fluctuation, it will be complicated and the point of the present study must be ambiguous.

Many of the statements contained in the discussion, although they seem correct, are pure speculation from the point of view of potential phenomena occurring during treatment - the Authors did not control the key parameters of the process, as well as what specific organisms developed.

     Indeed, our conclusion includes the speculation. Therefore, we usually used “suggest” for the speculation, and used “indicate” only when the results showed directly the phenomena. For the comment of the last part, we would like to give a same answer as described before.

This article is missing conclusions - I haven't found any conclusion chapter.

Conclusion was added at the last part.

The purpose of the research is not clear to me. It is well known that the smaller the bubbles, the better the air / oxygen mass transfer to the wastewater and it is crucial for biological treatment. The scientific novelty must be clearly marked in the article.

     For showing the purpose and the novelty of the research clearly, abstract was changed almost completely (l. 14-34), and some explanations were added to the Introduction (l. 72-97). In addition, some parts of the Discussion were also revised as relating to and emphasizing the novelty described in the Introduction.

Based on my comments and overall impression, I suggest major revision.

Round 2

Reviewer 1 Report

Reviewer 1:

I reviewed the revised version manuscript entitled"Effective Purification of Eutrophic Wastewater from the Beverage Industry by Microbubbles".The paper has been improved and can be accepted. I do not have further comments.

Reviewer 4 Report

This is my second review of this article. The Authors answered all of my questions. Suggested corrections have been applied. I suggest to accept this article in its present form.